# Impaired *in vitro* Interferon-γ production in patients with visceral leishmaniasis is improved by inhibition of PD1/PDL-1 ligation

**Yegnasew Takele[1,2,3], Emebet Adem[2¤a], Susanne Ursula Franssen[3¤b], Rebecca Womersley[1], Myrsini Kaforou[1], Michael Levin[1], Ingrid Müller[1], James Anthony Cotton[3], Pascale Kropf[1] ***

**1** Department of Infectious Disease, Imperial College London, London, United Kingdom, **2** Leishmaniasis Research and Treatment Centre, University of Gondar, Gondar, Ethiopia, **3** Wellcome Sanger Institute, Wellcome Genome Campus, Hinxton, United Kingdom

¤a Current address: University of Greenwich at Medway, Kent, United Kingdom
¤b Current address: Division of Evolutionary Biology, Faculty of Biology, LMU Munich, Germany
* p.kropf@imperial.ac.uk

**Data Availability Statement:** All relevant data are within the manuscript and its Supporting Information files.

## Abstract

Visceral leishmaniasis (VL) is a neglected tropical disease that causes substantial morbidity and mortality and is a growing health problem in Ethiopia, where this study took place. Most individuals infected with *Leishmania donovani* parasites will stay asymptomatic, but some develop VL that, if left untreated, is almost always fatal. This stage of the disease is associated with a profound immunosuppression, characterised by impaired production of Interferonγ (IFNγ), a cytokine that plays a key role in the control of *Leishmania* parasites, and high expression levels of an inhibitory receptor, programmed cell death 1 (PD1) on CD4+ T cells. Here, we tested the contribution of the interaction between the immune checkpoint PD1 and its ligand PDL-1 on the impaired production of IFNγ in VL patients. Our results show that in the blood of VL patients, not only CD4+, but also CD8+ T cells express high levels of PD1 at the time of VL diagnosis. Next, we identified PDL-1 expression on different monocyte subsets and neutrophils and show that PDL-1 levels were significantly increased in VL patients. PD1/PDL-1 inhibition resulted in significantly increased production of IFNγ, suggesting that therapy using immune checkpoint inhibitors might improve disease control in these patients.

## Author summary

Visceral leishmaniasis is a neglected tropical disease, that affects the poorest of the poor in low and middle-income countries. It is caused by a parasite, *Leishmania*, that is transmitted during the blood meal of an insect. When individuals cannot control *Leishmania* replication, they develop visceral leishmaniasis, that is characterised by enlarged spleen and liver, low blood cell counts and wasting. We have previously shown that lymphocytes from these patients have an impaired ability to produce a soluble mediator, IFNγ, that contribute to the killing of the parasites, but that this was restored after successful anti-

**Funding:** YT is funded by a Wellcome Trust Training Fellowship in Public Health and Tropical Medicine (204797/Z/16/Z). JAC is funded by Wellcome via core funding of the Wellcome Sanger Institute (grant 206194). MK is funded by a Wellcome Trust Sir Henry Wellcome Fellowship (206508/Z/17/Z). The funders had no role in study design, data collection and analysis, decision to publish, or preparation of the manuscript.

**Competing interests:** The authors have declared that no competing interests exist.

leishmanial treatment. Here we identified high expression levels of a marker, PD1, on lymphocyte; that has been associated with dysfunctional lymphocytes. We also identified the ligands of this marker, PDL1, on different blood cells. Furthermore, we showed that blocking the interaction between PD1 and PDL1 resulted in increased levels of IFNγ. These results suggest that treatment that blocks the interaction of PD1 with PDL1 might improve disease management and patient care.

## Introduction

Visceral leishmaniasis (VL) is one of the most neglected tropical diseases [1]. An estimated 550 million individuals are at risk of VL in high-burden countries: 17,082 new cases of VL were reported in 2018, with Brazil, Ethiopia, India, South Sudan and Sudan–which each reported more than 1000 VL cases–represented 83% of all cases globally in that year [2]. The remote location of VL endemic areas and the lack of surveillance make its likely that this is a significant underestimate of the real burden of VL in endemic areas. VL imposes a huge pressure on low and middle income countries and delays economic growth, with an approximate annual loss of 2.3 million disability-adjusted life years [3]. In Ethiopia, VL is caused by *Leishmania* (*L.*) *donovani* and is one of the most significant vector-borne diseases, with over 3.2 million people at risk of infection [4]. VL is a growing health problem, with spreading endemic areas and a steady increase in incidence since 2009 [5].

The majority of infected individuals control the parasite replication and do not progress to disease, they remain asymptomatic. In contrast, some individuals will progress and develop visceral leishmaniasis that is characterised by hepatosplenomegaly, fever, pancytopenia and wasting; this stage of the disease is generally fatal if left untreated [6,7]. One of the main immunological characteristic of VL patients is their profound immunosuppression [8]: these patients do not respond to the leishmanin skin test, their peripheral blood mononuclear cells (PBMCs) have an impaired capacity to produce IFNγ and proliferate in response to *Leishmania* antigen; this dysfunctional response to antigen challenge is restored following successful chemotherapy [9–11]. These findings show that T cell responses are impaired in VL patients, but the mechanisms leading to this impairment remain to be fully understood.

Using a whole blood assay (WBA), we have previously shown that whole blood cells from VL patients from Northern Ethiopia displayed an impaired capacity to produce IFNγ in response to stimulation with soluble *Leishmania* antigens (SLA) at time of diagnosis; but that these cells gradually regained their capacity to produce IFNγ over time after successful treatment [12,13].

We have recently shown that high levels of PD1 on CD4$^+$ T cells–an inhibitory receptor that can be expressed on exhausted T cells–was a hallmark of VL patients at time of diagnosis and that this was associated with low production of IFNγ [13]. The interaction of PD1 with its ligand PDL-1 contribute to T cell dysfunction [14]. Therefore, in this study we aimed to identify which cells express PDL-1; and determine whether the impaired production of IFNγ can be improved by interfering with the PD1/PDL1 pathway.

## Methods

### Ethics statement

This study was approved by the Institutional Review Board of the University of Gondar (IRB, reference O/V/P/RCS/05/1572/2017), the National Research Ethics Review Committee

(NRERC, reference 310/130/2018) and Imperial College Research Ethics Committee (ICREC 17SM480). Informed written consent was obtained from each patient and control.

## Subjects and sample collection

For this cross-sectional study, 10 healthy male non-endemic controls were recruited among the staff of the University of Gondar, Ethiopia, these individuals had not travelled to endemic areas and all tested negative by rk39; as well as 16 male patients with visceral leishmaniasis (VL patients) were recruited from the Leishmaniasis Treatment and Research Center, University of Gondar. The exclusion criterion was age <18 years. The diagnosis of VL was based on positive serology (rK39) and the presence of *Leishmania* amastigotes in spleen aspirates [15]. All patients were treated with a combination of sodium stibogluconate (20mg/kg body weight/day), and paromomycin (15mg/kg body weight/day) injections for 17 days according to the Guideline for Diagnosis and Prevention of Leishmaniasis in Ethiopia [16].

## Sample collection and processing

8ml of blood was collected in heparinised tubes and was used as follows: 3ml for the whole blood assay (WBA) and 5ml to purify PBMC and neutrophils as described in [17] for flow cytometry.

Four millilitres of blood (median white blood cells: $2.05 \pm 0.18 \times 10^6$ cells/ml) from 14 VL patients at time of diagnosis were collected in heparinised tubes and 4 x 1ml aliquots were distributed in 4 tubes and stimulated with SLA (10μg/ml) alone, SLA + anti-PD1 (Nivolumab, Pb3, BioVision, 1μg/ml), SLA + isotype control (QA16A15, Biolegend, 1 μg/ml) and with PBS as a negative control. The tubes were incubated for 24 hours at 37˚C, and supernatants were collected and stored at -20˚C until further analysis.

Soluble *Leishmania* antigen (SLA) was prepared using *Leishmania donovani* clinical isolates [18] collected from splenic aspirates of VL patients. To grow *Leishmania* parasites from spleen aspirations, the following culture media was used: 500ml of M199 medium (Sigma, USA) which was enriched with 25mM hepes, 0.2μM folic acid, 5ml vitamin mix, 1mM hemin, 1mM adenine, 800μM Biopterin, 5ml of Penicillin streptomycin and 50ml fetal bovine serum (Sigma, USA). Stationary-phase promastigotes were harvested and centrifuged at 4500 rpm for 20 minutes, the pellet was washed three times with cold PBS (Sigma, USA). The pellet was adjusted to $2 \times 10^9$/ml and resuspended in the following reagent: 50mM of EDTA, 50mM of HCL, 100mM of Phenylmethanesulfonyl fluoride and 5mg/ml of Leupeptin (Sigma, USA). The SLA suspension was sonicated 4–5 times for 15 seconds at 10 Hz and centrifuged at 27,000xg for 30 minutes at 4˚C. The lipid layer was removed from the surface of the supernatant. The remaining supernatant was ultra-centrifuged at 100,000xg for 4hrs at 4˚C. The supernatant was collected and was stored at -20˚C.

IFNγ levels were measured in the supernatant of the WBA using IFN gamma ELISA Kit (Invitrogen) according to the manufacturer's instructions.

Flow cytometry: the following antibodies were used directly *ex vivo*: CD4^FITC (OKT-4), CD8^PE CY7 (RPA-T8), PD1^PE (J105), PDL-1^PE(MIH1), CD15^APC (MMA) (eBioscience), CD14^APC (M5E2) and CD16^FITC (B73.1) (Biolegend) as described in [19].

Acquisition was performed using a BD Accuri C6 flow cytometer, at least 30,000 lymphocytes, 10,000 neutrophils and 5,000 monocytes were acquired, and data were analysed using BD Accuri C6 analysis software.

## Statistical analysis

Data were evaluated for statistical differences as specified in the legend of each figure. The following tests were used: Mann-Whitney or Wilcoxon tests. Differences were considered

statistically significant at $p<0.05$. Unless otherwise specified, results are expressed as median ±SEM. * = $p<0.05$, ** = $p<0.01$, *** = $p<0.001$ and **** = $p<0.0001$.

# Results

## Clinical and haematological parameters

The cohort of 16 VL patients and 10 healthy non-endemic controls that were recruited for this study were all male and aged-matched (Table 1). *Leishmania* amastigotes were present in all splenic aspirates from VL patients (parasite grade (+): 2.5±0.5). All VL patients presented with low BMI, fever, splenomegaly, and pancytopenia (Table 1).

## PD1 expression on T cells

Efficient effector functions of specific T cells are of crucial importance for the clinical outcome of visceral leishmaniasis. One of the main immunological features of VL patients in Ethiopia is their impaired ability at time of diagnosis (ToD) to produce antigen-specific IFNγ in a whole blood assay (WBA); after successful treatment the impaired IFNγ production is restored over time [12,19]. Our recent work [19] showed that the gradual increase in antigen-specific IFNγ production during follow-up is accompanied by a gradual decrease of PD1 expression on CD4+ T cells. Results presented in Fig 1A and 1B show that directly *ex vivo*, not only CD4+ T cells, but also CD8+ T cells expressed PD1. In both T cell subsets, the expression levels of PD1 on CD4+ T cells and CD8+ T cells were significantly higher as compared to controls ($p<0.0001$ and 0.0004, respectively).

## Monocytes and neutrophils express PDL-1

We have hypothesised that the high levels of PD1, via its interaction with PDL-1, is a major contributor of T cell hyporesponsiveness in VL patients [19]. However, the phenotypes of PDL-1 expressing cells have not yet been identified in VL patients.

Depending on the expression levels of CD14 and CD16, monocyte can be subdivided in three distinct subsets: classical (CD14+/CD16low), intermediate (CD14+/CD16+) and non-

**Table 1. Clinical and haematological parameters.**

|  | ToD | Controls | *p* values |
|---|---|---|---|
| **Clinical parameters** | | | |
| **Age (years)** | 25±2.1 | 26±2.2 | 0.5057 |
| **Parasite grade (+)** | 2.5±0.5 | nd | na |
| **BMI (kg/m$^2$)** | 16.2±0.4 | 21.0±1.0 | <0.0001 |
| **Fever (ºC)** | 37.4±0.2 | 36.0±0.1 | <0.0001 |
| **Spleen size (cm)** | 8.5±1.0 | 0.0±0.0 | <0.0001 |
| **Haematological parameters** | | | |
| **WBC (μl of blood x $10^3$)** | 1.8±0.1 | 5.8±0.7 | <0.0001 |
| **RBC (μl of blood x $10^6$)** | 3.0±0.2 | 5.4±0.2 | <0.0001 |
| **Platelets (μl of blood x $10^4$)** | 5.7±1.2 | 24.8±2.0 | <0.0001 |

A cohort of VL patients (n = 16) were recruited at time of diagnosis (ToD). Clinical and haematological parameters were assessed as described in Materials and Methods and compared to those of a cohort of age- and sex-matched non-endemic healthy controls (n = 10). Spleen sizes were measured below the left costal margin. nd = not done, na = not applicable.

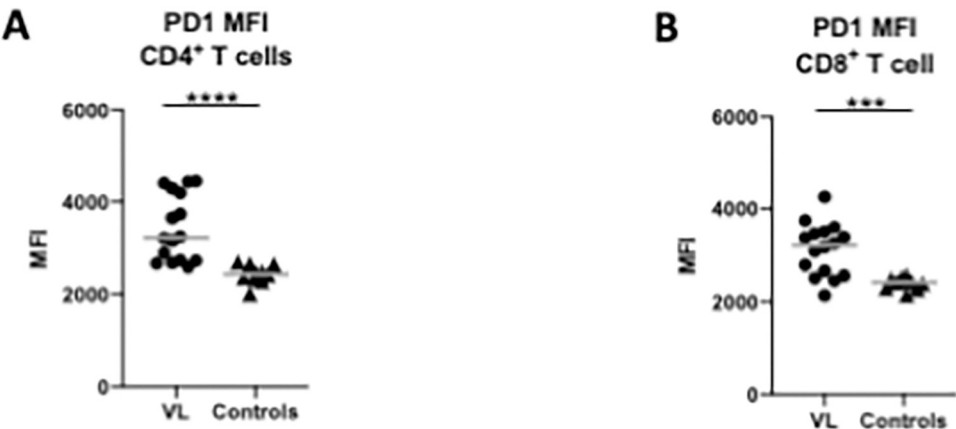

**Fig 1. Expression of PD1 on T cells.** The Median Fluorescence Intensity (MFI) of PD1 was measured *ex vivo* by flow cytometry on CD4+ T cells (**A**) and CD8+ T cells (**B**) in the PBMCs from VL patients (n = 16) and controls (n = 10). PBMCs were isolated from whole blood as described in Material and Methods. The gating strategy is detailed in S1 Fig. Statistical differences were determined by a Mann-Whitney test. Each symbol represents the value for one individual, the straight lines represent the median.

classical (CD14$^{low}$/CD16$^+$). Each subset can display different functions: broadly, classical monocytes exhibit strong phagocytosis abilities; intermediate monocytes are characterised by their abilities to induce T cell stimulation and high ROS production, as well as pro-angiogenic abilities; and non-classical monocytes are characterised by their patrolling behaviour of the vascular endothelium [20]. Importantly, the role of these different subsets in VL patients is poorly characterised [21].

Here we tested whether the different monocyte subsets isolated from VL patients express PDL-1, and if so, if they express it differentially. Results presented in Fig 2 and S1 Table show that directly *ex vivo*, monocytes in the PBMCs of VL patients express PDL1. All three monocyte subsets: classical (CD14$^+$/CD16$^{low}$, Fig 2A), intermediate (CD14$^+$/CD16$^+$, Fig 2B) and non-classical (CD14$^{low}$/CD16$^+$, Fig 2C) expressed significantly more PDL-1 at ToD compared to controls ($p$>0.0001, $p$>0.0001 and $p$ = 0.0061, respectively).

Next, we assessed if neutrophils also expressed PDL-1 at ToD. Results presented in Fig 2D show that directly *ex vivo*, neutrophils expressed significantly higher levels of PDL-1 at ToD than controls ($p$ = 0.0022).

We also tested for correlations between the parasite grades and the PD1 and PDL-1 MFI, but none of these correlations were significant ($p$>0.05).

## Interfering with the PD1/PDL-1 pathway results in increased production of IFNγ

Based on the increased expression levels of PD1 on T cells and PDL-1 on monocytes and neutrophils and the low levels of IFNγ produced in the WBA [19], we tested if blockade of the PD1/PDL-1 interaction improves IFNγ production. Results presented in Fig 3A show that interfering with the PD1/PDL-1 ligation resulted in significantly higher levels of IFNγ (p = 0.0006). As expected, there was no significant difference between the levels of IFNγ produced in response to SLA alone and those produced in response to SLA in the presence of an isotype control (Fig 3B).

These results suggest that the impaired ability of whole blood cells to produce IFNγ efficiently can be improved by PD1/PDL-1 blockade.

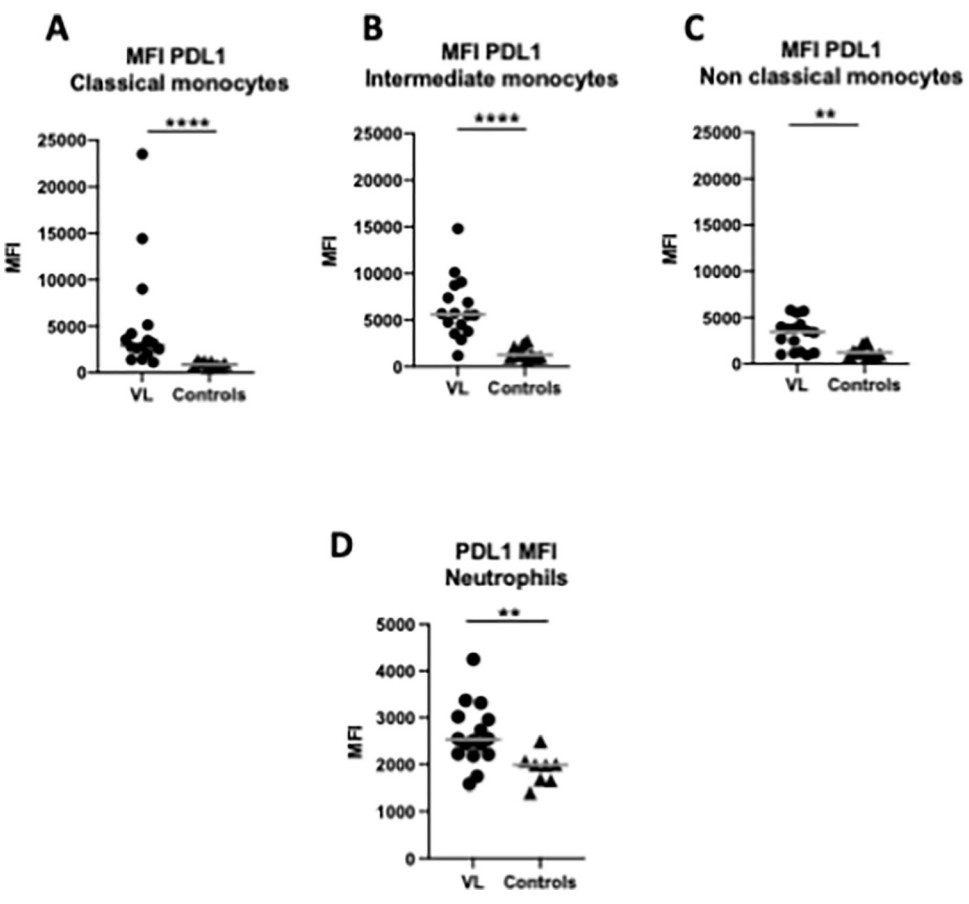

**Fig 2. Monocytes and neutrophils express PDL-1.** *Ex vivo* PDL-1 MFI was measured by flow cytometry on monocytes from PBMCs of VL patients n = 16) and controls (n = 10). PBMCs were isolated from whole blood as described in Material and Methods. The gating strategy is detailed in S2 Fig. **A.** PDL-1 expression on classical monocytes (CD14+/CD16low). **B.** PDL-1 expression on intermediate monocytes (CD14+/CD16+). **C.** PDL-1 expression on non classical monocytes (CD14low/CD16+). Statistical differences were determined by a Mann-Whitney test. **D.** *Ex vivo* PDL-1 MFI was measured by flow cytometry on neutrophils in the PBMCs from VL patients (n = 16) and controls (n = 8). Neutrophils were purified from whole blood as described in Materials and Methods. The gating strategy is detailed in S3 Fig. Statistical differences were determined by a Mann-Whitney test. Each symbol represents the value for one individual, the straight lines represent the median.

## Discussion

Severe immune suppression has been previously documented in VL patients [9], however, it is still poorly understood. Here we show that at the time of diagnosis of VL both CD4+ and CD8+ T cells express high levels of PD1. The interaction of inhibitory receptors such as PD1 with their ligand play a crucial role in controlling autoreactivity and immunopathology. These receptors are also upregulated during T cell activation, but this is transient. In contrast, chronic stimulation of T cells results in the maintenance of high levels of PD1 expression on T cells; the duration and degree of this chronic stimulation are key to T cell exhaustion and dysfunction [22]. VL patients present at the Leishmaniasis Treatment and Research Center with severe disease and on average have had VL symptoms for around 2 months [19]. It is therefore likely that the chronic stimulation by *Leishmania* antigen plays a major role in the maintenance of high expression levels of PD1 on T cells in these patients. Several inflammatory mediators have been shown to result in the upregulation of PDL-1 on different cell types such as neutrophils and monocytes [23]. It is well established that high levels of inflammation are

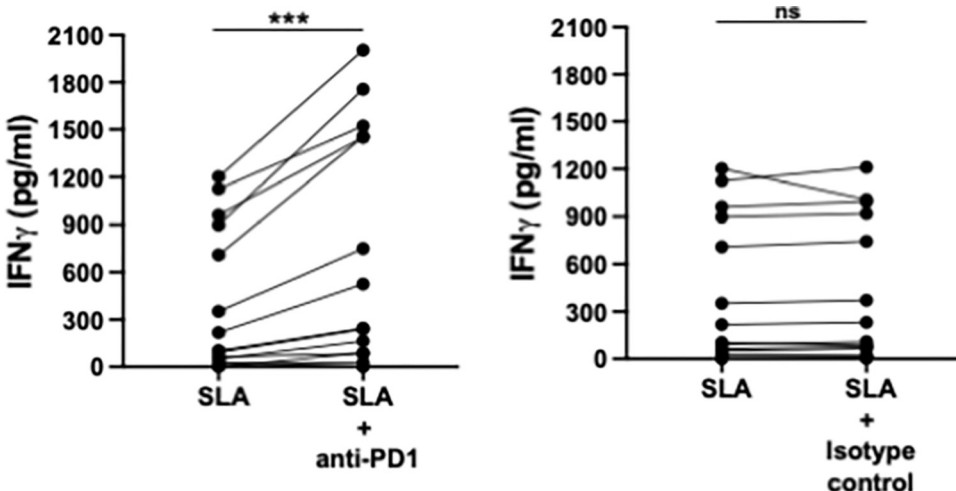

**Fig 3. Interfering with the PD1/PDL-1 pathway results in increased production of IFNγ.** Whole blood cells from VL patients at ToD (n = 14) were stimulated with **A.** SLA in the presence (1μg) or absence of anti-PD-1 mAb; or **B.** SLA in the presence (1μg) or absence of an isotype control. IFNγ was measured by ELISA in the supernatant after 24hrs. Statistical differences were determined by a Wilcoxon test. Each symbol represents the value for one individual.

common in VL patients [9] and indeed, we found high levels of TNFα, IL-6, IL-8, IFNγ [19], as well as IL-17 [24] in the plasma of VL patients at the time of diagnosis. IL-10 has also been shown to upregulate PDL-1 on monocytes [25] and indeed, high levels of IL-10 in plasma are a hallmark of VL patients [9,19]. Furthermore, IFNγ is also a central driver of PDL-1 expression [26]; and indeed, our results have shown that IFNγ is high at ToD in VL patients. Therefore, we propose that in VL patients, the chronic inflammation and antigenic stimulation results in the upregulation of both PD1 and PDL-1, that results in T cell exhaustion; that is manifested in the whole blood assay by impaired production of IFNγ. Exhausted T cells become hypofunctional, they maintain some of their effector functions such proliferation [14,22,27]. Exhausted T cells have been shown to contribute to the control of chronic infections and limit immunopathology [14,22,27]. It is therefore possible that during the acute state of infection, exhausted T cells might still contribute to the control of parasite replication, as well as limit tissue damage.

In addition to inflammatory mediators, parasites can upregulate PDL-1 on immune cells. A study showed that *Leishmania amazonensis*, a parasite causing cutaneous leishmaniasis, can upregulate PDL-1 on both mouse and human neutrophils, these neutrophils have the ability to suppress the production of IFNγ by CD8[+] T cells [28].

Our data show that the levels of PD1 expression of CD4[+] and CD8[+] T cells and PDL-1 expression on monocytes and neutrophils are significantly increased at ToD as compared to control. The controls in our study were all individuals living in Gondar, an area of Ethiopia that is not endemic for VL, they had not travelled to an endemic area and they all tested negative for rk39. However, we cannot exclude that these individuals might have been infected with *L. donovani* and were asymptomatic.

Whereas PDL-2 is another co-signalling molecule that can bind to PD1 and contribute to T cell suppression, our knowledge of its functional role is still poor [29]. Since we did not test for the expression levels of PDL-2 on monocytes and neutrophils, we cannot exclude that it contributes to further T cell suppression in VL patients; and that additional blockade of PDL-2/PD1 pathway might have further improved the production of IFNγ in the whole blood assay. Indeed, a study in a hamster model showed that interfering with the PDL-2 pathway resulted in decrease in parasite burden [30].

PD-L1/PD1 interaction is a powerful antagonist of TCR signal transduction, as well as CD28 and ICOS costimulatory signalling. This interaction results in impaired cytokine production and cell cycle arrest; as well as in the reduction of the transcription of Bcl-X$_L$, a mitochondria transmembrane molecule with anti-apoptotic properties. The ligation of PDL-1 and PD1 abrogates the phosphorylation of phosphorylation of various signaling molecules, such as ERK, Vav, PLCγ, and PI3K [31,32]. A recent study showed that T cell suppression via PD-1 ligation is a result of the inactivation CD28 signaling by PD1 ligation [33]. However, the mechanism of PD-1-mediated T cell inhibition are still poorly understood.

In cutaneous leishmaniasis, it was also shown that blocking PD-1 resulted in increased production of IFNγ by circulating T cells [34]. The authors speculate that PD-1 blockade may result in improved T cell effector functions and thereby reduce the pathology. Furthermore, in a mouse model of infection with *Leishmania amazonensis*, mice treated with anti-PD1 and anti-PDL-1 displayed larger lesions, that contained significantly lower parasite [35].

Our results that show that interfering with the PD1/PDL-1 pathway results in increased production of IFNγ disagree with the results presented by Gautam *et al.* [36]. This study was performed in India, where VL patients present at a significantly earlier stage of the disease, with less severe symptoms; and indeed, the results of their study show that the levels of IFNγ measured in the whole blood assay were not impaired at time of diagnosis as compared to those detected after successful treatment [37]. As previously discussed, [12,19], we propose that the impaired production of IFNγ we measured in the WBA from the Ethiopian patients is closely related to the fact that they present late, often in a critical state. Of note, we cannot exclude that the parameters we measured in the blood in the present study might be different in other organs affected by VL, such as spleen, bone marrow or lymph nodes.

In Ethiopia, the first line of treatment against visceral leishmaniasis is a combination therapy of Sodium Stibogluconate and Paromomycin [16]. While this treatment shows a good safety profile and a good efficacy, there are still severe side effects, such as cardiotoxicity and nephrotoxicity [38]. In our latest study, four individuals from our cohort of 50 VL patients died during treatment [19]. PD1/PDL-1 blockade has emerged as a front-line treatment for several types of cancer [39]. However, little is known about its potential use in chronic infectious diseases. In a non-human primate model of simian immunodeficiency virus blockade of the PD1 pathway improved T cell effector functions and resulted in more efficient viral control [40]. Paradoxically, in a 3D cell culture model of tuberculosis despite the increased levels of both PD1 and PDL-1, blockade of PD1 promoted the replication of *M. tuberculosis* [41]. Several studies in experimental models have suggested that immune checkpoint blockade maybe relevant to treat several infectious diseases [42]. Even though immune checkpoint blockade can cause organ-specific immune-related adverse events, such as hepatitis and colitis, as well as systemic inflammation [43]; further studies are needed to determine whether blockade of the PD1/PDL1 pathway can be used to improve therapies against infectious diseases. In the case of visceral leishmaniasis, it might be particularly useful in tackling the severe form of the disease, to prevent the treatments' adverse side effects, by allowing the use of shorter courses or reduced doses of current anti-leishmanial treatments.

## Supporting information

**S1 Table. PDL-1 MFI was measured *ex vivo* by flow cytometry on monocytes from PBMCs of VL patients n = 16) and controls (n = 10).** PBMCs were isolated from whole blood as described in Material and Methods. The gating strategy is detailed in S2 Fig. Statistical differences were determined by a Mann-Whitney test.
(DOCX)

**S2 Table. PDL-1 MFI was measured *ex vivo* by flow cytometry on monocytes from PBMCs of VL patients n = 16) and controls (n = 10).** PBMCs were isolated from whole blood as described in Material and Methods. The gating strategy is detailed in S2 Fig. Statistical differences between the 3 different subsets were determined by a Kruskal-Wallis test.
(DOCX)

**S1 Fig. Flow cytometry analysis of PD1 expression on CD4$^+$ and CD8$^+$ T cells.** PBMCs were purified as described in Materials and Methods and the expression levels (Median Fluorescence Intensity [MFI]) of PD1 on CD4$^+$ and CD8$^+$ T cells were measured by flow cytometry. A. FSC and SSC of the lymphocyte gate (P1). B. CD4$^+$ T cells in the lymphocyte gate (P1). C. CD8$^+$ T cells in the lymphocyte gate (P1). D. PD1 (M2) on CD4$^+$ T cells gate (M7). E. PD1 (M3) on CD8$^+$ T cells gate (M1).
(TIFF)

**S2 Fig. Flow cytometry analysis of PDL-1 expression on Classical CD14$^{++}$CD16$^-$, Intermediate CD14$^{++}$CD16$^+$ and Non-classical CD14$^+$CD16$^{++}$ Monocytes.** PBMCs were purified as described in Materials and Methods. PBMCs were stained with anti-human CD14$^{APC}$, CD16$^{FITC}$ and PDL-1$^{PE}$ and the expression levels (Median Fluorescence Intensity [MFI]) of PDL-1 on the three subsets of monocytes were measured by flow cytometry. **A**. FSC and SSC of the monocyte gate (P4). **B**. Different monocyte subsets based on the expression levels of CD14 and CD16: classical (R18), Intermediate (R19) and Non-classical (R20) monocytes. **C, D** and **E**. PDL-1 MFI on the Classical (M31), Intermediate (M32) and Non-classical monocytes (M33).
(TIFF)

**S3 Fig. Flow cytometry analysis of PDL-1 expression on neutrophils.** Neutrophils were purified as described in Materials and Methods and the expression of PDL-1 on neutrophils was measured by flow cytometry. **A**.FSC and SSC of the neutrophil gate (P3). **B**. CD15$^+$ neutrophils in P3. **C**. PDL-1 MFI (M10) on neutrophils in M1.
(TIFF)

## Acknowledgments

We are grateful to the staff of the Leishmaniasis Research and Treatment Centre for their support and DNDi for supporting the VL treatment service at the University of Gondar.

## Author Contributions

**Conceptualization:** Yegnasew Takele, Ingrid Müller, Pascale Kropf.

**Formal analysis:** Yegnasew Takele, Emebet Adem, Susanne Ursula Franssen, Rebecca Womersley, Myrsini Kaforou, Michael Levin, Ingrid Müller, James Anthony Cotton, Pascale Kropf.

**Funding acquisition:** Yegnasew Takele, Pascale Kropf.

**Investigation:** Yegnasew Takele, Emebet Adem, Ingrid Müller, Pascale Kropf.

**Writing – original draft:** Yegnasew Takele, Ingrid Müller, James Anthony Cotton, Pascale Kropf.

**Writing – review & editing:** Yegnasew Takele, Emebet Adem, Susanne Ursula Franssen, Rebecca Womersley, Myrsini Kaforou, Michael Levin, Ingrid Müller, James Anthony Cotton, Pascale Kropf.

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
