## [Decision Letter · Decision Letter 0]

18 Jan 2022

Dear Dr. Kropf,

Thank you very much for submitting your manuscript "Impaired in vitro Interferon-γ production in patients with visceral leishmaniasis is improved by inhibition of PD1/PDL-1 ligation" for consideration at PLOS Neglected Tropical Diseases. As with all papers reviewed by the journal, your manuscript was reviewed by members of the editorial board and by several independent reviewers. In light of the reviews (below this email), we would like to invite the resubmission of a significantly-revised version that takes into account the reviewers' comments. 

The reviewers raise a number of methodological issues that must be addressed. The gating strategies for flow cytometry must be provided. The role of neutrophils in PD-1 mediated IFNg suppression should be addressed. Thee effect of PD1 blockade on individual cell populations (intracellular cytokines) should be addressed.

We cannot make any decision about publication until we have seen the revised manuscript and your response to the reviewers' comments. Your revised manuscript is also likely to be sent to reviewers for further evaluation.

Sincerely,

Peter C. Melby, M.D.

Associate Editor

Charles Jaffe

Deputy Editor

The reviewers raise a number of methodological issues that must be addressed. The gating strategies for flow cytometry must be provided. The role of neutrophils in PD-1 mediated IFNg suppression should be addressed. Thee effect of PD1 blockade on individual cell populations (intracellular cytokines) should be addressed.

Reviewer's Responses to Questions

**Key Review Criteria Required for Acceptance?**

**Methods**

-Are the objectives of the study clearly articulated with a clear testable hypothesis stated?

-Is the study design appropriate to address the stated objectives?

-Is the population clearly described and appropriate for the hypothesis being tested?

-Is the sample size sufficient to ensure adequate power to address the hypothesis being tested?

-Were correct statistical analysis used to support conclusions?

-Are there concerns about ethical or regulatory requirements being met?

Reviewer #1: The objectives of the study were clearly described and articulate with the hypothesis of the study. The work was designed in a way to address the objectives and the population was clearly described and is appropriate to the aim of the investigation. The sample size is sufficient to test the hypothesis of the work. The statistical analysis was appropriate. The study protocol was approved by research ethics committees.

Reviewer #2: The authors had previously shown that in VL patients, CD4+ T cells from whole blood stimulated with SLA express high levels of PD-1, and that this is associated with a reduced secretion of IFN-gamma.

In the current study, the authors aim to determine which cells produce PD-L1, the ligand of PD-1, and if the reduction in IFN-gamma secretion can be abolished by cellular treatment with the PD-1 blocker Nivolumab.

Overall, the study provides conclusive results on the stated objectives, and we are aware that the number of experiments that can be conducted with precious patient’s material (8ml of whole blood) is limited. However, the amount of data shown to prove the link between the single observations is a little sparse. 

With 16 VL patients and 10 control individuals, the sample size seems reasonable to reveal data for answering the main questions. The statistical analysis of the data is appropriate.

Reviewer #3: About Figure 1: Are T cells stimulated? Is the difference of PD-1 expression in the presence or absence of SLA?

About Figures 2 and 3: are they stimulated with SLA? Are there diferrences?

About figure 4: It is missing isotype control. 

It is missing the production of SLA

**Results**

-Does the analysis presented match the analysis plan?

-Are the results clearly and completely presented?

-Are the figures (Tables, Images) of sufficient quality for clarity?

Reviewer #1: The analysis presented by the author match with the analysis plan, and the results are clearly presented. The figures are of sufficient quality.

Reviewer #2: The experiments in this study are well chosen to provide information on the objectives.

Table 1 summarizes the clinical and haematological parameters of the VL patients and the non-infected control individuals.

Could the authors distinguish between healthy individuals (not infected until then) and individuals that might have had an asymptomatic Leishmania infection before?

It might be beneficial to graphically depict the parasite grade of the different infected individuals for better insight into the variance in parasite load, which might correlate with the extent of the later analyzed PD-1 and PD-L1 expression as well as the IFN-y secretion.

Fig. 1: Besides the already previously published PD-1 expression on CD4+ T cells (A), the authors now additionally determine PD-1 expression on CD8+ T cells (B), which is significantly increased in comparison to the healthy controls. 

Data in Panel 1a was already published previously, so why show it again, new donors? Please specify. 

Please show the gating strategy that was applied to measure the PD-1 MFI of the T cell subpopulations. Did the authors set a gate on cell size (FSC/SSC), single cells, did they check for cell viability? Information should be provided in the material and methods section or graphically in the results section. 

Fig. 2: As the binding of PD-L1 to PD-1 can elicit a T cell exhaustion phenotype which might explain a pronounced disease development. Thus, the authors measured the relative amount of PD-L1 that is expressed on blood monocytes.

For all three examined monocyte subtypes, there is a significant difference between PD-L1 expression in healthy controls and VL patients. 

However, it’s not clearly stated why exactly those three monocyte subtypes were chosen for the analysis. What is their individual role in the recognition of Leishmania parasites or in activation of the adaptive immune system?

Can the authors confirm that these blood monocytes were (partially) infected with Leishmania? If not, what would be the trigger to make them express PD-L1. How would they get in contact with the parasites?

Would it be an option to additionally investigate PD-L1 expression on macrophages from splenic aspirates, or even more specific from infected macrophages?

Figure 2D and 2E are redundant to Fig 2A-C as this is just another combination of the graphs. This is not correct, please change accordingly and put all 6 datasets into one chart in order to show all relevant comparisons and significances.

Fig. 3: Here the authors show the PD-L1 expression by neutrophils. As in Fig 2, it is not clear if these neutrophils are (partially) infected with Leishmania in the blood, which would explain their activation. Once recruited to the affected organs (spleen, liver) neutrophils might fulfill their function as local drivers of an immune response but they might not get back into the blood stream.

The authors should also state in the figure legend or results text which surface markers were used for identifying the cells as neutrophils (CD15, CD16)

Fig.4: Treatment of whole blood cells from VL patients with SLA in the presence of the anti-PD-1 mAB Nivolumab increases IFN-gamma secretion as shown by ELISA.

How much SLA was used (concentration) and from which strain was it produced, please specify.

According to the methods section, cells were treated with 1µg Nivolumab. What was the total volume of the assay and/or the (approx.) number of cells treated?

The authors should also provide data on whole blood cells derived from healthy patients in this assay.

Reviewer #3: Figure 1: Please include the representative dot plot

Figure 2: Please include the representative dot plot

What is the difference in figure 2D? The median is very similar. 

Figure 3: Please include the representative dot plot

Figure 4: 

Is there a possibility to do a correlation of expression of PD1 versus IFNg production?

What is the level of PD-1 before and after the treatment with anti-PD1.

**Conclusions**

-Are the conclusions supported by the data presented?

-Are the limitations of analysis clearly described?

-Do the authors discuss how these data can be helpful to advance our understanding of the topic under study?

-Is public health relevance addressed?

Reviewer #1: The conclusion are supported by the data provided. Yet, study limitations are not so well described. The subject is of highly public health relevance and the authors have clearly emphasized the importance of the current study.

Reviewer #2: no

no

yes

yes

Reviewer #3: It is missing intracellular citokines IFNg on CD4 and CD8 T cells to see the impact of anti-PD1 in each population.

**Editorial and Data Presentation Modifications?**

Reviewer #1: Minor revision.

Reviewer #2: -

Reviewer #3: Figure 2 and 3 could be combined.

**Summary and General Comments**

Reviewer #1: The authors in this study investigated the expression levels of PD-1 on CD4 and CD8 T cells of visceral leishmaniasis patients, and of its ligand PDL-1 on monocytes and neutrophils. The authors have found that anti-PD-1 treatment of whole blood cells of VL patients increases the production of INFg.

General comments:

The article requires minor modification that can be addressed through revision. The study addressed the interruption of PD-1/PDL-1 pathway, but do not mention the other ligand PDL-2 which has been shown to contribute as well to the success of anti-PD-1 treatment. It would be interesting to introduce PDL-2 as another ligand of PD-1 and discuss whether PDL-2 has been addressed in VL or not and the consequences.

Methods:

- How SLA is produced? Could the authors describe on that as well?

- What are the clones of the monoclonal antibodies used for FACS? This information should be in the paper.

- What is the clone of the commercial anti-PD-1 used for blocking assays? Is this information possible to add in the paper?

- The authors inform that acquisition was performed on a BD Accuri B6, how many events were acquired per sample? How was the gating strategy used for the analyses?

- Why the authors did not use CD3 or CD45 in the panels to better define the cell types investigated?

- The authors should explain why they did not use the same antibody isotype of PD-1 as control for the antibody blocking assay. This is a critical part of the methods used in the paper.

Figures:

Table 1. should be reformatting.

Fig 1. It would be interesting to provide all the gating strategy used to define CD4 and CD8 T cells, and the other cell types. This information could be as supplementary material to the article. Did the authors observed a frequency difference as well? Do the authors imagine that these T cells are exhausted? It would be interesting to discuss this as well.

The information on the markers to define each cell type evaluated should be in figure legends as well.

Fig 2. Do the authors have any data on the levels of PDL-1 of in situ macrophages from biopsies? It would be interesting for the discussion.

Fig 4. Do the authors have an idea if blocking affects the production of other cytokines that could be relevant for VL as well? I understand that the main hypothesis was drafted on the impairment of IFNg production ToD, but it would be scientifically relevant to show the data on other cytokines as well, such as IL-10, IL-4, IL-2 and TNF.

Discussion:

The authors could at least discuss how the interaction of PD-1/PDL-1 mechanistically contribute to IFNg impairment during VL.

Reviewer #2: -

Reviewer #3: It is missing important references

- PD-1 Blockade Modulates Functional Activities of Exhausted-Like T Cell in Patients With Cutaneous Leishmaniasis

DOI: 10.3389/fimmu.2021.632667

- Leishmania Parasites Drive PD-L1 Expression in Mice and Human Neutrophils With Suppressor Capacity

DOI: 10.3389/fimmu.2021.598943

- Immunotherapy using anti-PD-1 and anti-PD-L1 in Leishmania amazonensis-infected BALB/c mice reduce parasite load

DOI: 10.1038/s41598-019-56336-8

PLOS authors have the option to publish the peer review history of their article (what does this mean?). If published, this will include your full peer review and any attached files.

Reviewer #1: Yes: Rafael de Freitas e Silva

Reviewer #2: No

Reviewer #3: No
---

## [Decision Letter · Decision Letter 1]

30 May 2022

Dear Dr. Kropf,

We are pleased to inform you that your manuscript 'Impaired in vitro Interferon-γ production in patients with visceral leishmaniasis is improved by inhibition of PD1/PDL-1 ligation' has been provisionally accepted for publication in PLOS Neglected Tropical Diseases.

Best regards,

Peter C. Melby, M.D.

Associate Editor

Charles Jaffe

Deputy Editor

The authors have sufficiently addressed the concerns of the reviewers

Reviewer's Responses to Questions

**Key Review Criteria Required for Acceptance?**

**Methods**

-Are the objectives of the study clearly articulated with a clear testable hypothesis stated?

-Is the study design appropriate to address the stated objectives?

-Is the population clearly described and appropriate for the hypothesis being tested?

-Is the sample size sufficient to ensure adequate power to address the hypothesis being tested?

-Were correct statistical analysis used to support conclusions?

-Are there concerns about ethical or regulatory requirements being met?

Reviewer #1: The objectives of the current work articulate with the hypothesis and the authors have chosen the correct tools to address it. In this new version the population studied was appropriately described and the number of samples is enough to draw the conclusions. It seems that proper statistics has been employed and there are no ethical concerns in this study.

**Results**

-Does the analysis presented match the analysis plan?

-Are the results clearly and completely presented?

-Are the figures (Tables, Images) of sufficient quality for clarity?

Reviewer #1: In this work the analysis presented matched with the analysis planned. The results were clearly and completely presented and the figures of sufficient quality.

**Conclusions**

-Are the conclusions supported by the data presented?

-Are the limitations of analysis clearly described?

-Do the authors discuss how these data can be helpful to advance our understanding of the topic under study?

-Is public health relevance addressed?

Reviewer #1: The conclusions are supported by the data presented and the limitations or questions have been clarified in this new version and in the rebuttal letter. The authors discussed how their data can advance the understanding of the role of PD1/PDL1 pathway during visceral leishmaniasis. The topic of this study is of utmost importance in worldwide public health.

**Editorial and Data Presentation Modifications?**

Reviewer #1: Accept.

**Summary and General Comments**

Reviewer #1: The data presented in this work is really important to understand the role of PD1/PDL1 during T cell disfunction/exhaustion in visceral leishmaniasis. So far, VL is a public health issue in many countries and the current drugs available are limited and the ones available are extremely toxic. I believe the study addresses a new perspective in the field. Yet, it lacks a more comprehensive panel of effector molecules involved in T cell disfunction and how interruption of PD1/PDL1 also impact on the production of other molecules and genes. Even tough the study is extremely significant for the field of parasite immunology and also helps paving the way for new treatments against VL.

PLOS authors have the option to publish the peer review history of their article (what does this mean?). If published, this will include your full peer review and any attached files.

Reviewer #1: **Yes: **Rafael de Freitas e Silva

---

## [Editor Report · Acceptance letter]

20 Jun 2022

Dear Dr. Kropf,

We are delighted to inform you that your manuscript, "Impaired in vitro Interferon-γ production in patients with visceral leishmaniasis is improved by inhibition of PD1/PDL-1 ligation," has been formally accepted for publication in PLOS Neglected Tropical Diseases.

Best regards,

Shaden Kamhawi

co-Editor-in-Chief

Paul Brindley

co-Editor-in-Chief
